# The French Telematic Magazine *Art Accès* (1984–1987)

## Marie Vicet

Research Unit HAR Histoire des Arts et des Représentations, Université Paris Nanterre, 92000 Nanterre, France; marie.vicet@gmail.com

**Abstract:** Created in 1984 by the French artists ORLAN, Frédéric Develay and Frédéric Martin and shown for the first time at the Centre Pompidou during the exhibition *Les Immatériaux* (28 March to 15 July 1985), the telematic magazine *Art Accès* has marked the history of the art on Minitel, the French Videotex system in use between 1980 and 2012. For ORLAN and Frédéric Develay, *Art Accès* was a way both to propose an artistic and cultural alternative to a purely utilitarian and mercantile content, but also to explore the possibilities of a 'poor' medium. Working within the framework of the magazine, ORLAN and Frédéric Develay invited visual artists, but also poets and musicians to use videotex, to transgress it in all possible ways and thus to make an original work that is made by this medium and for this medium. Although the French Minitel network ended in 2012 and the magazine has long since disappeared, there are still traces, fragments or documents that allow us to reconstruct its history. This essay proposes an initial study of this telematic experience and of some of its most emblematic creations.

**Keywords:** telematic art; minitel; videotex system; Frédéric Develay; ORLAN; *Les Immatériaux*

## 1. Introduction

Created in 1984 by the French artists ORLAN, Frédéric Develay and Frédéric Martin and shown for the first time at the Centre Pompidou during the exhibition *Les Immatériaux* (28 March to 15 July 1985), the telematic magazine *Art Accès* has marked the history of the art on Minitel.[1] Minitel is the French acronym for "Médium interactif par numérisation d'information téléphonique" [interactive medium by digitisation of telephone information] and refers to the connection terminal of the French Videotex system called Télétel in use between 1980 and 2012. This service, which allowed users to consult pages composed of text and graphics on a Minitel terminal connected to the telephone network, thus combined computing and telecommunications. It was seen by many as one of the forerunners of the Internet as we know it today, See (Mailland and Driscoll 2017) and (Schafer and Thierry 2012).

Before the online magazine *Art Accès*, other artists had already seized the Minitel to carry out artistic experiments. The first of these was the artist Fred Forest with his participative installation *Bourse de l'imaginaire* at the Centre Georges Pompidou in June 1982, where various communication equipment (telephones, answering machines, fax machines, etc.) were made available to the public, including Minitel terminals, while the service was still in an experimental phase. The following year, the first novel on Minitel named *ASCOO* (for "Abandon Commande Sur Ordre Opérateur" ["Command Abandonment On Operator Order"]) was presented by a group of young graphic designers made up of Jacques-Élie Chabert, Camille Philibert, Jean-René Bader and the journalist Jean-Paul Martin at the exhibition *Electra* organised at the Museum of Modern Art of the City of Paris (10 December 1983 to 5 February 1984). Thus, from 1982 and over a little more than a decade, some thirty works were produced for the Minitel network (Bureaud 2016, p. 139). The magazine *Art Accès* is the most ambitious art project on Minitel, bringing together more than a hundred artists from the fields of art, experimental literature and music over a period of three years. However, since the cessation of the Télétel network on 30 June 2012, the Minitel as a telecommunications network no longer exists and the data it contained has

disappeared with it.[2] Today, therefore, only a very small glimpse of what the magazine was like remains thanks to the paper archives preserved by Frédéric Develay and recently acquired by the Kandinsky Library of the Centre Georges Pompidou.[3] They allow us to get some idea of the richness and great diversity of the contributions made to *Art Accès*. This essay proposes an initial study of this telematic experience and of some of its most emblematic creations. This history is still very incomplete and deserves further research.

## 2. Creation Context

In the history of telecommunications, the Minitel occupies a special place in France. Conceived under French President Valéry Giscard d'Estaing in 1978, the Minitel was launched by the Ministry of Posts, Telegraphs and Telephones (PTT) in 1983 with an online directory, with the aim of replacing the paper telephone directory in the long term. It was then opened in 1984 to private operators who could offer their own services (messaging, classified ads, banking, games, etc.). In an effort to make it accessible to the general public and with the aim of "computerising" French society,[4] a terminal was provided free of charge to every household (Schafer and Thierry 2012, p. 10). This is what made the Minitel so successful with French users. This initiative sets France apart from other countries with a comparable service where users paid for the rental of the terminal. As a result, few individuals in these countries (UK, Germany and USA) subscribed to the Minitel-like service.[5]

Art with videotext has existed in various countries (notably Canada and Brazil) but it is in France that it has developed the most (Bureaud 2016, p. 140). With the widespread deployment of the Minitel throughout the country, some artists saw the possibility of reaching a new public directly from their homes. In 1985, this was also the argument put forward by Frédéric Develay and ORLAN in their choice to use this new medium as a creative platform: "France is one of the most advanced countries in the field of telematics, especially if we consider the number of Minitel terminals currently distributed: more than 1 million, and soon many more (it is announced that there will be 2 million by the end of 1985)" (Develay and ORLAN 1986, pp. 39–41). They thought they could potentially reach more than two million "readers", or even more with the growing number of households equipping themselves with a terminal during the 1980s. These two artists, interested in new technologies, decided to create in 1984 "the first telematic magazine of contemporary art" (Develay and ORLAN 1987, p. 57). The magazine was divided into three sections: the first—and by far the most important—was devoted to "Visual Arts", the second to "Literature" and the third to "Music" (See Figure 1). The art critic Annick Bureaud analyses:

> With a medium that was considered, in today's words, as "innovative" and a vector of "future for society", for ORLAN and Develay, it was crucial to propose a cultural and an artistic alternative to its purely utilitarian and economical use and to raise and explore conceptual issues by challenging its limitations, confronting the then-shiny bland computer images associated with progress, as well as established art forms.
>
> (Bureaud 2016, p. 144)

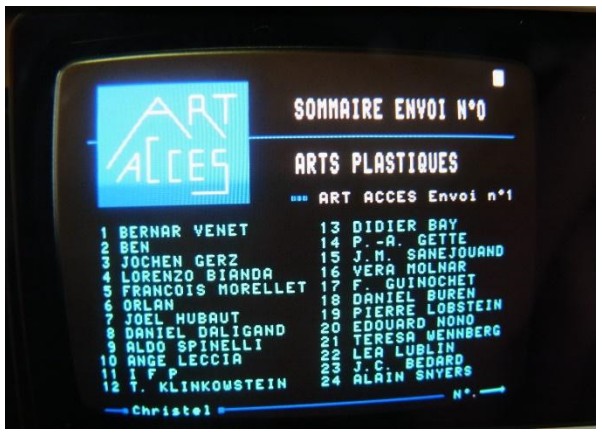

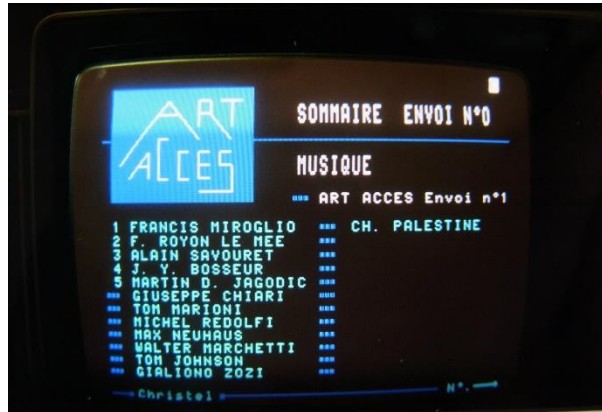

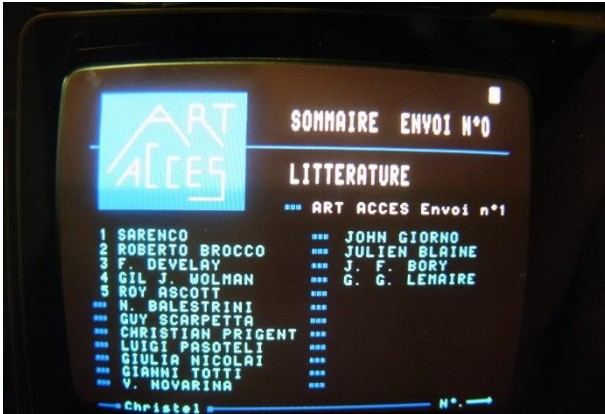

**Figure 1.** Images of the summaries of the three sections "Visual Arts", "Literature" and "Music" of the *Art Accès* issue no. 0. Collection Frédéric Develay © Kandinsky Library, MNAM/CCI, Centre Pompidou, photos: Andreas Broeckmann.

### 3. The Constraints of Creation on Minitel

Although ORLAN and Frédéric Develay chose Minitel to create their online art magazine, it was far from being a tool that was conducive to artistic creation. The terminal distributed to users from 1982 onwards was a low-tech system consisting of a keyboard and a 9-inch cathode-ray screen (23 cm diagonal) capable of displaying characters in alpha mosaic mode. The equipment had rather limited graphic capabilities: "8 colours or 8 levels of grey, frames, animation allowed by scanning, flashing, masking or dynamic formatting. The screen allows the display of 25 rows of 40 alpha numeric and graphic characters, i.e., 127 available signs" (Schafer and Thierry 2012, pp. 50–51). The keyboard was initially composed of 57 keys, including nine function keys whose use was not always clear to users:

the "Summary", "Cancel", "Back", "Repeat", "Guide", "Correction", "Continue", "Send", "Connect/End" keys (Schafer and Thierry 2012, p. 74).

Each participant in the magazine received four pages of instructions listing what can and cannot be done with the Minitel (the alphabetical, numerical, alphabetical, mosaic and colour elements that can be used), how the pages are displayed, and a screen grid based on the composition of a Minitel screen, consisting of 40 squares horizontally by 23 vertically. Each square was rectangular and was divided into six squares. The pages were created in colour on a composition keyboard (eight colours were possible: black, white, yellow, red, blue, green, cyan and magenta), but at home users had access almost exclusively to black and white terminals.[6] The colours were then translated into eight levels of grey (See (Develay and ORLAN 1986, p. 39)). It was with these constraints that artists had to deal with a technology that was not designed for drawing, had a low resolution image on the screen and had no sound. However, during demonstrations, exhibitions or presentations, the magazine was shown in colour. "In this case, a Minitel terminal with a SCART socket is used to connect it to a TV screen or a colour Minitel," as explained in the document provided to participants (See Figure 2).

**Figure 2.** *Cont.*

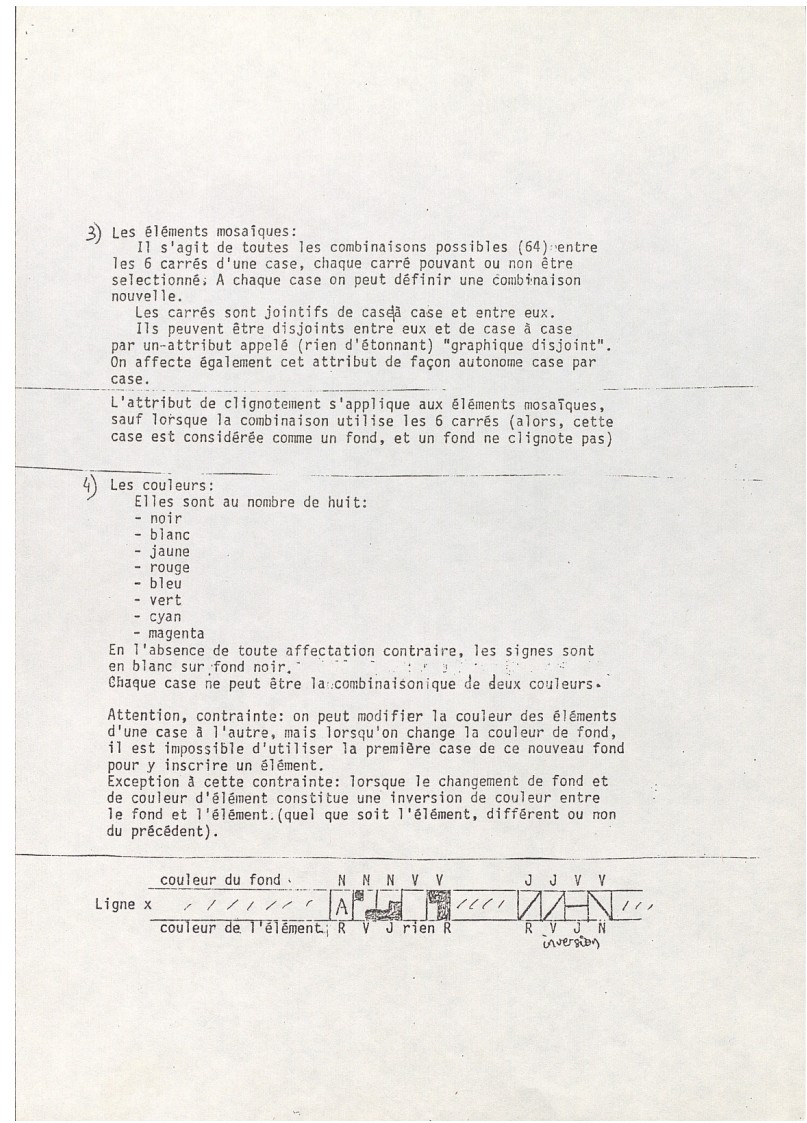

3) Les éléments mosaïques:
   Il s'agit de toutes les combinaisons possibles (64) entre
les 6 carrés d'une case, chaque carré pouvant ou non être
selectionné. A chaque case on peut définir une combinaison
nouvelle.
   Les carrés sont jointifs de case à case et entre eux.
   Ils peuvent être disjoints entre eux et de case à case
par un attribut appelé (rien d'étonnant) "graphique disjoint".
On affecte également cet attribut de façon autonome case par
case.

   L'attribut de clignotement s'applique aux éléments mosaïques,
sauf lorsque la combinaison utilise les 6 carrés (alors, cette
case est considérée comme un fond, et un fond ne clignote pas)

4) Les couleurs:
   Elles sont au nombre de huit:
      - noir
      - blanc
      - jaune
      - rouge
      - bleu
      - vert
      - cyan
      - magenta
En l'absence de toute affectation contraire, les signes sont
en blanc sur fond noir.
Chaque case ne peut être la combinaisonique de deux couleurs.

Attention, contrainte: on peut modifier la couleur des éléments
d'une case à l'autre, mais lorsqu'on change la couleur de fond,
il est impossible d'utiliser la première case de ce nouveau fond
pour y inscrire un élément.
Exception à cette contrainte: lorsque le changement de fond et
de couleur d'élément constitue une inversion de couleur entre
le fond et l'élément.(quel que soit l'élément, différent ou non
du précédent).

**Figure 2.** *Cont.*

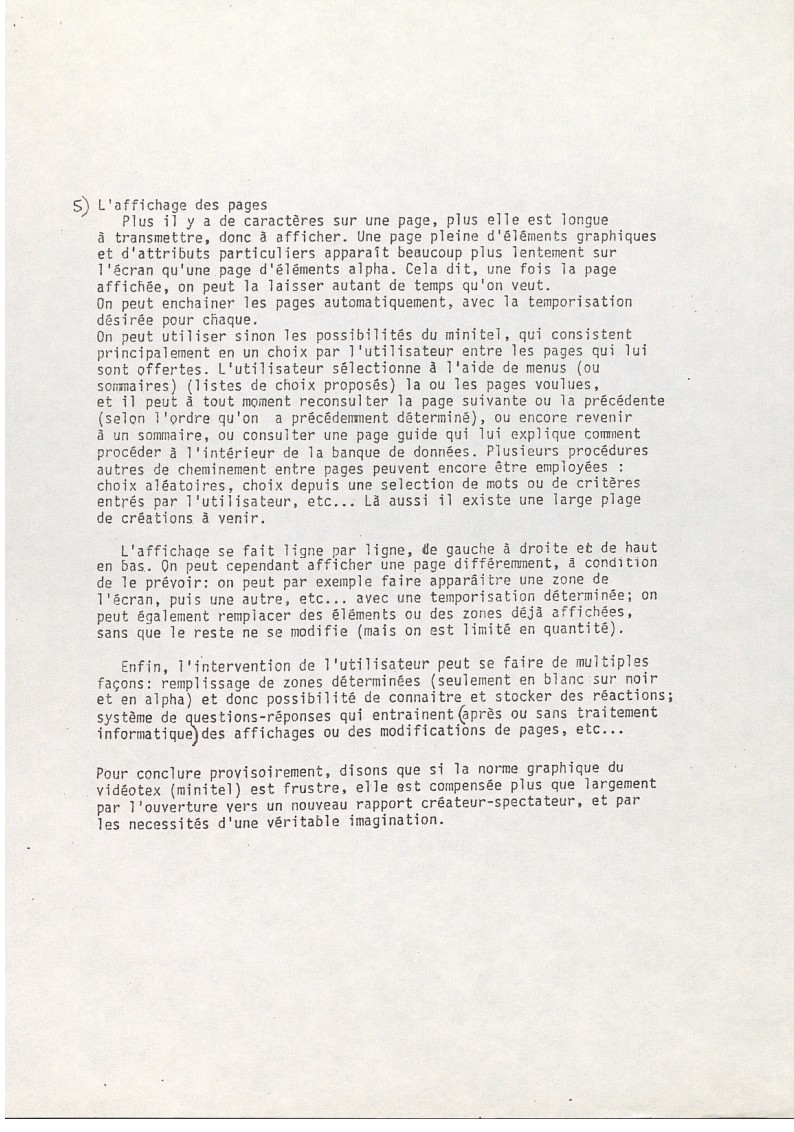

5) L'affichage des pages

Plus il y a de caractères sur une page, plus elle est longue à transmettre, donc à afficher. Une page pleine d'éléments graphiques et d'attributs particuliers apparaît beaucoup plus lentement sur l'écran qu'une page d'éléments alpha. Cela dit, une fois la page affichée, on peut la laisser autant de temps qu'on veut.
On peut enchainer les pages automatiquement, avec la temporisation désirée pour chaque.
On peut utiliser sinon les possibilités du minitel, qui consistent principalement en un choix par l'utilisateur entre les pages qui lui sont offertes. L'utilisateur sélectionne à l'aide de menus (ou sommaires) (listes de choix proposés) la ou les pages voulues, et il peut à tout moment reconsulter la page suivante ou la précédente (selon l'ordre qu'on a précédemment déterminé), ou encore revenir à un sommaire, ou consulter une page guide qui lui explique comment procéder à l'intérieur de la banque de données. Plusieurs procédures autres de cheminement entre pages peuvent encore être employées : choix aléatoires, choix depuis une selection de mots ou de critères entrés par l'utilisateur, etc... Là aussi il existe une large plage de créations à venir.

L'affichage se fait ligne par ligne, de gauche à droite et de haut en bas. On peut cependant afficher une page différemment, à condition de le prévoir: on peut par exemple faire apparaître une zone de l'écran, puis une autre, etc... avec une temporisation déterminée; on peut également remplacer des éléments ou des zones déjà affichées, sans que le reste ne se modifie (mais on est limité en quantité).

Enfin, l'intervention de l'utilisateur peut se faire de multiples façons: remplissage de zones déterminées (seulement en blanc sur noir et en alpha) et donc possibilité de connaître et stocker des réactions; système de questions-réponses qui entrainent (après ou sans traitement informatique) des affichages ou des modifications de pages, etc...

Pour conclure provisoirement, disons que si la norme graphique du vidéotex (minitel) est frustre, elle est compensée plus que largement par l'ouverture vers un nouveau rapport créateur-spectateur, et par les necessités d'une véritable imagination.

**Figure 2.** *Cont.*

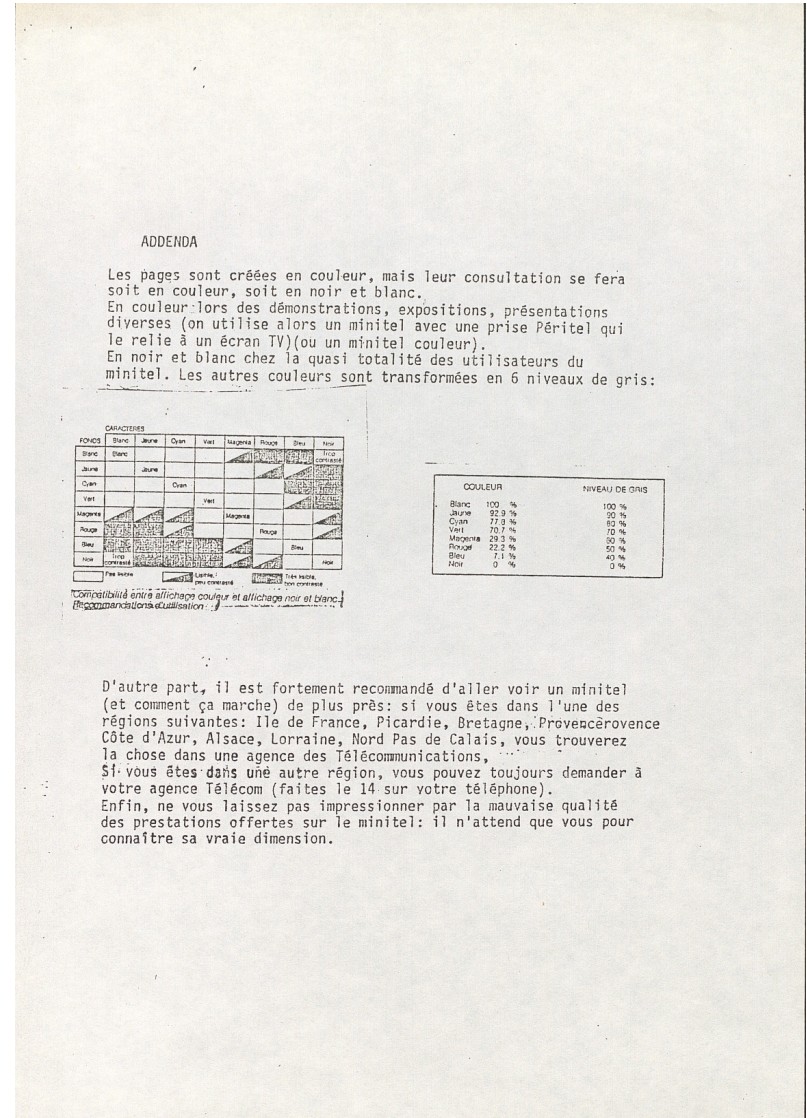

**Figure 2.** Document with instructions sent to each artist participating in *Art Accès*. Collection Frédéric Develay © Kandinsky Library, MNAM/CCI, Centre Pompidou.

It should be noted that if the contributors received a paper model of the screen grid in addition to the written instructions, it was because most of them, and in particular the foreign artists, sent their proposal by mail and thus drew on paper the succession of the screen pages of their project. In the vast majority of cases, the artists who participated in this magazine were not familiar with videotex technology and were unable to program the screen pages of their project themselves. The use of a technician was therefore essential. The technician tried to interpret the drawings sent by the artists and to adapt them to the constraints of the screen and the Minitel technology. When possible, the artists worked directly with him to create the screen pages of their project together (See Figure 3).

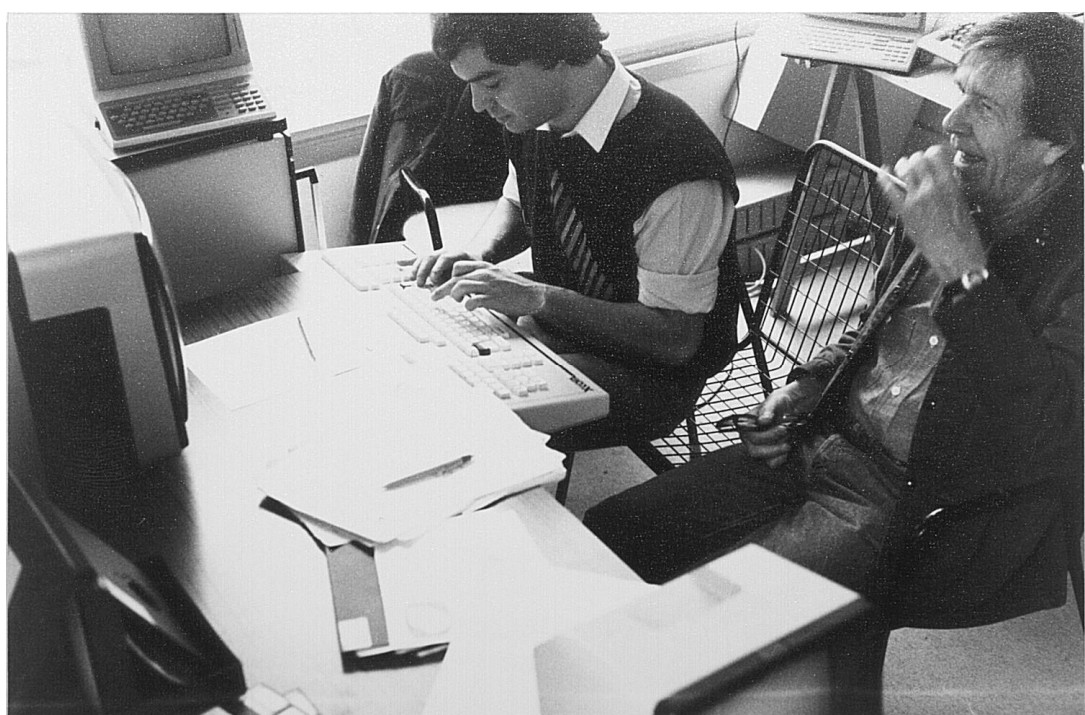

**Figure 3.** John Cage and the technician of *Art Accès* using a composition keyboard to create the screen pages of Cage's project "Two Ackus". Collection Frédéric Develay © Kandinsky Library, MNAM/CCI, Centre Pompidou.

The aim of the magazine was to propose original works "absolutely without concession [and] without thinking about the *general public*" (Develay and ORLAN 1987, p. 58) on one or more pages specially designed for the Minitel and to be seen in this context. Although the magazine's creators were demanding with regard to the artists' proposals, these were accompanied by a mediation to allow a better understanding. Before the work, the reader had access to a text written by the artist himself on one or more screen pages explaining his approach, how and why he has decided to create a work for the Minitel. After the work itself, another text, this time entrusted to an art critic chosen by the artist, presented the latter's work in greater detail. Users accessed these different pages by pressing the "next" key on the Minitel terminal keyboard. ORLAN explained in 1987: "We made two bets: both the bet to make works that were not for the general public and the bet to put a lot of text and a lot of images in this Minitel, which is an open tap and in which until now there has been very little cultural relevance apart from this great thing called messaging" (Develay and ORLAN 1987, p. 61).

## 4. Some Examples of Artistic Creations

Of the many contributions made during the three years of the magazine's existence, only a few traces remain, and only for certain projects: essentially screen photographs, preparatory drawings and correspondence. While they give us some idea of what the artist's project might have been, these secondary documents are far from fully reflecting it. The photographs are still images that do not show the animations that could be seen on the screen. Therefore, we cannot know with certainty the number of screen pages that a project contained, nor the order in which these pages were sequenced. As for the preparatory drawings, if they are not accompanied by screen photographs of the same project, we cannot know exactly how these were adapted for the Minitel screen and if they were indeed adapted.

We now propose to study a few projects whose archives we have been able to consult and which seem to us to be representative, to a certain extent, of the production of *Art Accès*.

*4.1. Visual Arts*

As far as the visual artists are concerned, it is worth noting that the choice was made in part for artists working in situ (such as Daniel Buren or Jochen Gerz) or with constraints (including Vera Molnar and François Morellet). The aim was for their works to play with and divert the constraints of the medium and, in some cases, to go against what was usually found on Minitel. Often, the artists' contributions are part of their plastic research and are an adaptation or reinterpretation of one or more of their works for the Minitel. *Art Accès* allowed them to experiment with a new medium but also to introduce their work to a new audience.

For his participation in *Art Accès*, the French artist Bernar Venet adapted his four series *Angles*, *Arcs*, *Diagonals* and *Indeterminate Lines* from 1976 in which he explores and redefines the line. His paintings and sculptures in wood and rolled steel became simple lines on the Minitel screen. Furthermore, from screen to screen, the reader saw a plastic typology unfold around the line, each variation being accompanied by its name: "Diagonals", "Angles", "Arcs" and "Indeterminate lines". For the Minitel, the artist reduced his plastic vocabulary to the essential.

The artist Daniel Buren, famous for his black and white stripes, has also been able to adapt and play with the constraints of the Minitel. In *Pochade no. 5* (See Figure 4), the only contribution without an accompanying text, the screen was initially completely black, allowing the artist's stripes to appear over the pages, before becoming completely black again and then giving way to a variation of squares of black and white lines and stripes. In the following screens, the artist played with the different levels of grey that the transition from colour to black and white brought to the screen. Thus, the white of his stripes was transformed into a gradation of grey in a series of screens. The indications below the image, such as "White + cyan = 100% + 77.8% = 100% + 80% Minitel grey", allowed readers to become aware of the programming carried out on the composition keyboard to achieve the result they were looking at.

For ORLAN, her participation in *Art Accès* was an opportunity to once again stage the character of Saint Orlan, which she had created in 1974 for the series of performances and plastic works *Le Drapé, le Baroque* (1974–1990) in which she drew inspiration from the figures of Madonnas, virgins, martyrs and saints of Christian iconography, and in particular those of Saint Theresa of Avila and Lisieux. On the Minitel screen, she was shown wearing a religious habit revealing a naked breast, reinterpreting the photograph *Étude documentaire: Le Drapé-Le Baroque, Sein Unique, Monstration Phallique* (1983).[7] The following screens showed only her naked breast (See Figure 5): "from screen to screen, one could zoom on the bared breast to reveal first the word "art" at the teat and then the words "new" (in English) and "vieux" (meaning "old")" (Bureaud 2016, p. 144). This was ORLAN's way of signifying that the new Minitel art had replaced the old one. The arrows with which the martyred saints were once pierced now take on the appearance of graphic signs. Here, ORLAN reveals herself as the new icon of telematic art, both religious and erotic.

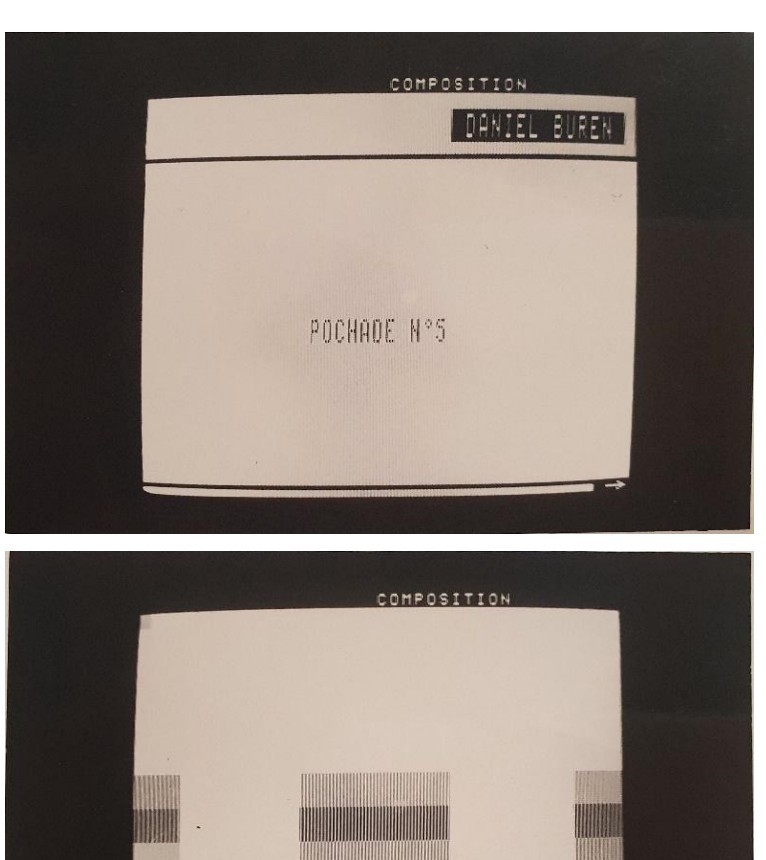

**Figure 4.** Images from "Pochade no. 5", project by the artist Daniel Buren for *Art Accès*. Collection Frédéric Develay © Kandinsky Library, MNAM/CCI, Centre Pompidou.

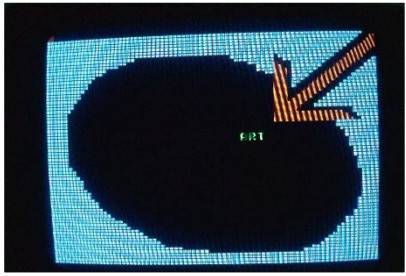 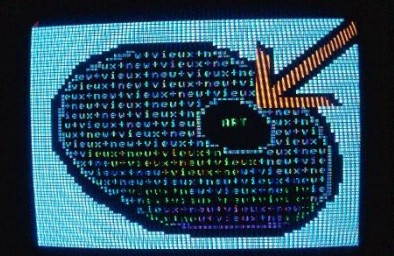

**Figure 5.** Images from "Sainte-ORLAN-au-sein-provocant", project by the artist ORLAN for *Art Accès*. Collection *Les Immatériaux* © Kandinsky Library, MNAM/CCI, Centre Pompidou.

For *Art Accès*, François Morellet chose to show a successive reinterpretation of three of his works on the Minitel screen. In a telegraphic style, he explains in the text accompanying his proposal: "I didn't respect the rules of the game. [ . . . ] I have only made three of my works assaulted by this medium, the exemplary killer of all works of art. I am delighted with the result. The main thing: my systems and my lack of interest in the medium have been well transmitted. Thank you Minitel."[8] As he points out, he was not interested in the support and therefore the medium of his works. Adapting them for the Minitel did not seem to him to be a denaturation of his works, although the result produced was

quite different from them. The first work he chose was the painting *16 carrés*[9] made in 1953, which he described as "the most minimalist work [he had] ever done" (Ida Biard 1985, p. 53). It consists of 16 white squares painted on an 80 cm wooden support and separated by black lines. This grid-work was the starting point for many of his plastic researches and has taken different forms. In 1964, he "neonised" and "cynetised" it for the Labyrinth of the GRAV (Groupe de Recherche d'Art Visuel) [Visual Art Research Group].[10] In 1985, he "minitelised" it for *Art Accès*. The two other works he chose to use for his intervention were *Trame 5° placée horizontalement*[11], dating from 1976, and *Géométree no. 92*[12] from 1985. While the result of *Trame 5° placée horizontalement* for the Minitel—although far from the original—respected the plastic form of its model, it was more complicated for *Géométree no. 92*. The work is not just a painting but includes, like all the works in this series, a tree branch and thus becomes three-dimensional. However, it was not possible with the Minitel to reproduce the thickness and the relief, only the shape of the branch on the canvas was rendered. It was therefore very difficult to recognize the original work without any indication. On the Minitel screen, the result appeared very impoverished.

Other artists imitated the services found on the Minitel with irony in their work. This was notably the case of the artists Alain Snyers and Lefèvre Jean-Claude or even Jean-François Bory in the literature (Develay and ORLAN 1986, p. 39). Alain Snyers created a fake news service for *Art Accès*, which he called "Infaux Service" (See Figure 6), a humorous parody of the ads usually found on Minitel. Users can read a succession of false announcements about the pedestrianisation of the Champs Élysées at certain times of the day, a bar offering free cocktails to the job seekers, or the financing of churches through the sale of their works of art. In a sociological art approach, Snyers used the medium to propose "a diversion of plausible urban facts", as he writes in the text accompanying his work.

For his part, the artist Lefèvre Jean-Claude offered the "Lefèvre Jean-Claude Archives 1971–1983" service (See Figure 7). Since the 1970s, he has been developing an approach centred on archiving and writing. And among his various projects, *LJC Archives* (started in 1983) brings together a set of documents that he began to collect in 1971. It includes leaflets, invitation cards, posters, texts and correspondence. "Like all archives, *LJC Archives* is the subject of an inventory that brings together a written description, in the form of lists, of the material and the environment of the exhibitions in which he participated or the projects that were not carried out" (Fleury 2009). For the artist, the Minitel was a place to disseminate his project by communicating the list of available files and indicating a telephone number to call for the public who wished to make an appointment to consult them. Ten years before the Internet was developed in France, he created the equivalent of his first website.

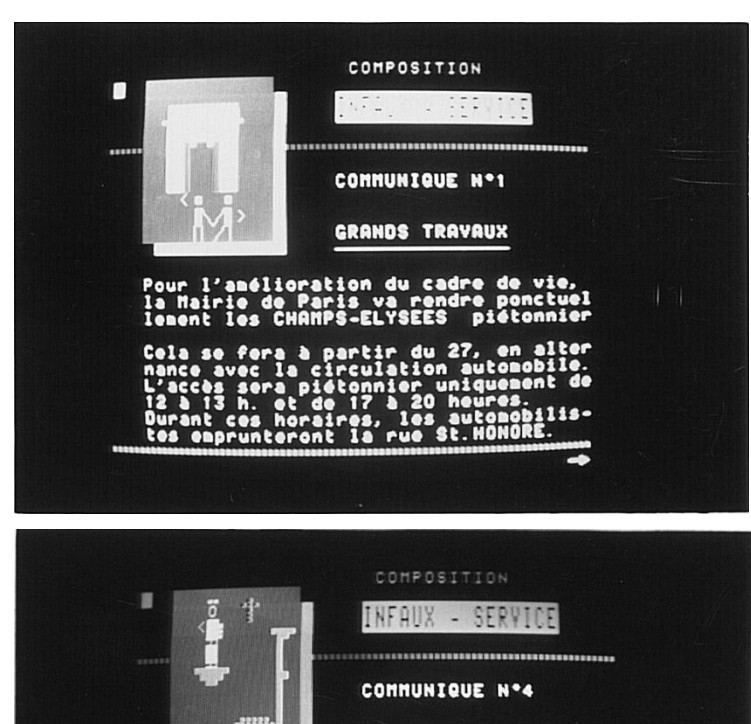

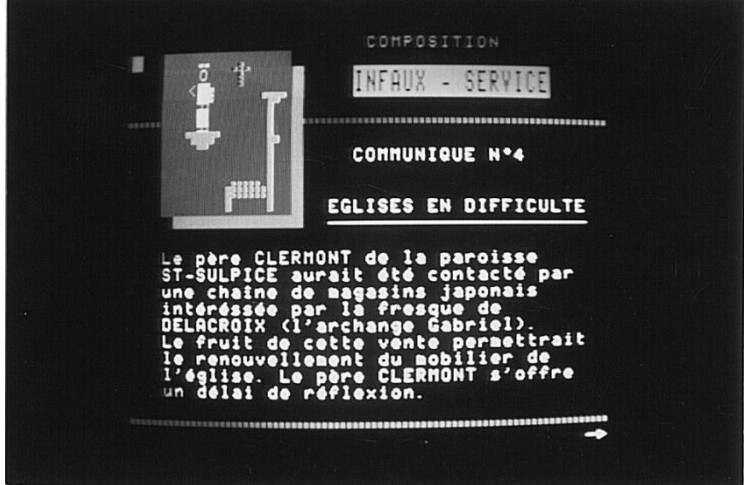

**Figure 6.** Images from "Infaux Service", project by the artist Alain Snyers for *Art Accès*. Collection Frédéric Develay © Kandinsky Library, MNAM/CCI, Centre Pompidou.

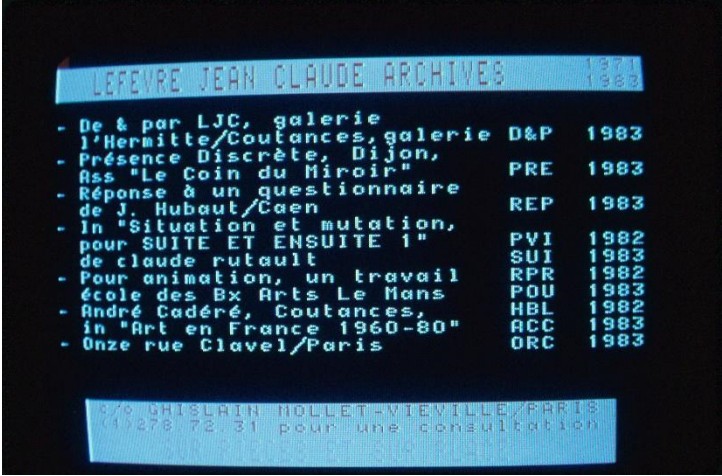

**Figure 7.** Image from "Lefèvre Jean-Claude Archives 1971–1983", project by the artist Lefèvre Jean-Claude for *Art Accès*. Collection *Les Immatériaux* © Kandinsky Library, MNAM/CCI, Centre Pompidou.

*4.2. Literature*

For the "Literature" section, the choice was made for poets or authors of experimental literature who were interested in new forms of writing and reading. The Minitel has provided them with a new medium for experimentation and a new channel of distribution, as Annick Bureaud explains:

> The Minitel welcomed not only hypertext graphic stories but also a myriad of forms of experimental and digital literature and poetry. For the most part, these creative works took place in the framework of *ART ACCES Revue* with works by Frédéric Develay, Tibor Papp, Philippe Bootz, Julien Blaine, Henri Chopin, Isidore Isou, and Jean-François Bory, to name but a few.
>
> (Bureaud 2016, p. 143)

Few traces remain of this literary production on Minitel. The challenge was for the authors' proposal to be anchored in their work but also to question the possibilities or constraints of the medium. However, as Frédéric Develay points out, the results and attitudes vary from one author to another, depending on whether or not they were used to using Minitel: "If we want to talk more specifically about the presence of writers and poets, the answers are quite different from those of visual artists and composers, namely that we find much less flexibility among writers, even among poets. [ ... ] The results are very diverse: they range from the simple transfer of the written word on paper to a screen to works that consider the plastic or computer possibilities of the medium (e.g., the work of Tibor Papp)." (Develay and ORLAN 1987, pp. 57–58)

While, as Francis Debyser deplores, many writers were unable to break away from paper and adapt to the Minitel (Anis and Lebrave 1987, p. 72), Frédéric Develay worked on what he called the "paper fatigue" and experimented with different media and modes of writing (ORLAN 1987, pp. 65–66). In his proposal for *Art Accès* entitled *L'Écrire/Lire*, he played with words and with videotex and mixed the letters of the words "l'écrire" and "lire" and made them appear on the Minitel screen intermittently by flashing.

*4.3. Music*

With regard to interventions in the musical field, it is important and interesting to note that the Minitel was a medium without sound and therefore only visual. Composers and musicians have turned to the creation of new kinds of scores or have revisited the codes of musical notation. "Musicians who play on the plasticity of unplayable scores have been chosen (Jean-Yves Bosseur, Alain Savouret, Frank Royon Le Mée, Martin Davori Jagodic, Pierre Mariétan, Michel Redolfi, Max Neuhaus)" (Develay and ORLAN 1986, p. 39), list the creators of the magazine. In his proposal entitled *Mosaïque 7* (See Figure 8), the composer Francis Miroglio, known for combining pictorial art and musical art in his practice, but also for the invention of an intuitive musical notation, created a system for the Minitel made up of graphic signs where each sign symbolising a family of instruments is linked to those indicating the type of sound to be produced and the tempo to be respected. From screen to screen, the score was revealed to the musicians who wished to interpret it.

The French composer Frank Royon Le Mée created for *Art Accès* a "music of the eyes, that of silence for imaginative ears in front of images without sound, forms without voice" as he wrote in the text preceding his work. He conceived three pages of music for the Minitel that once again the user had to interpret to create his own "musical moment" (See Figure 9). It was up to him to find the tempo, the tonality and the timbre, whereas on the screen the score created by Royon Le Mée was no more evident than Miroglio's to decode. However, the user was free to read and interpret it as he wished in order to create his own concert.

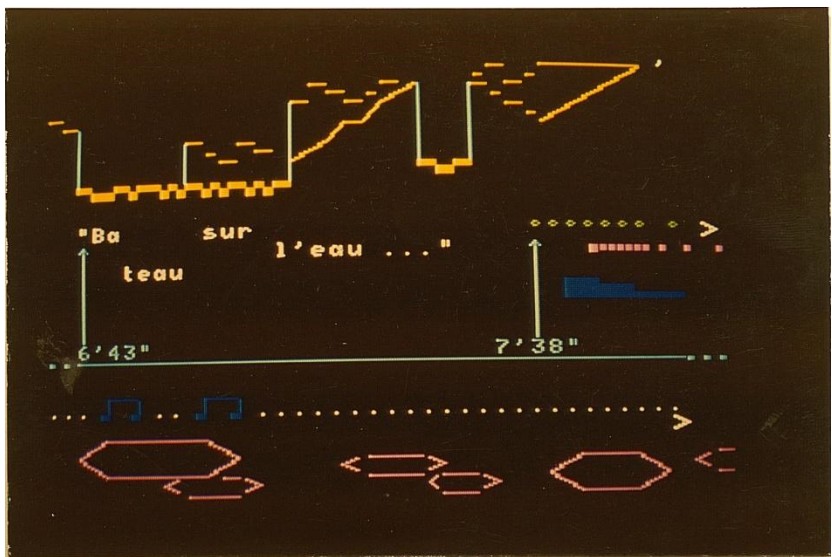

**Figure 8.** Image from "Mosaïque 7", project by the composer Francis Miroglio for *Art Accès*. Collection Frédéric Develay © Kandinsky Library, MNAM/CCI, Centre Pompidou.

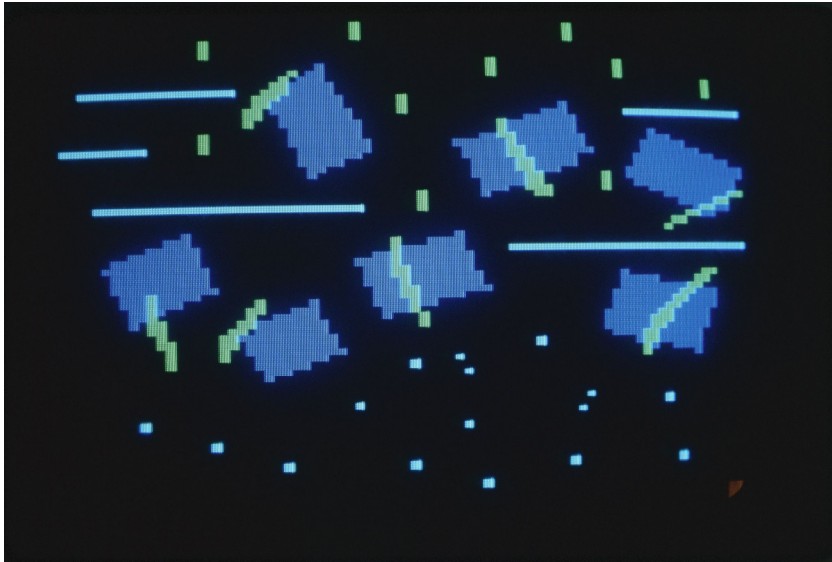

**Figure 9.** Image from "Une musique pour un petit écran ?", project by the composer Frank Royon Le Mée for *Art Accès*. Collection *Les Immatériaux* © Kandinsky Library, MNAM/CCI, Centre Pompidou.

## 5. Exhibition of the Magazine

In addition to being directly accessible from home on any Minitel terminal using the code 3615 AL33,[13] the magazine was regularly shown at exhibitions and artistic events during its existence. Its presentations have taken several forms. During the exhibition *Les Immatériaux* at the Centre Pompidou, the magazine was available on a single Minitel terminal among others in the site *Labyrinthe du langage*, which brought together numerous artistic projects on Minitel and computer terminals.[14] The same year, at the Locarno International Video Art Festival,[15] the magazine was presented in the form of an installation made up of forty Minitel terminals of various models topped by a television set retransmitting the image of ORLAN's Madonna (See Figure 10). Each terminal was switched on and showed a creation by one of the magazine's artists. This was made possible by one of the possibilities offered by the Minitel terminal: "by making two quick CONNECTION/END, you can keep a small work by Buren or John Cage as a luminous painting for five minutes or fifteen

years" ([Anis and Lebrave 1987](), p. 71), explained ORLAN in 1987. It was the Minitel as a screen and electronic board that was highlighted in this type of presentation, not the telecommunications network. The sculptural aspect was favoured over the navigation between the projects and the numerous screen pages.[16] At the FIAC (International Contemporary Art Fair of Paris) in 1985,[17] the magazine was also presented as an installation with 18 Minitel terminals displayed in rows of three on a shelf, while a terminal allowed the public to consult the magazine freely (See Figure [11]).

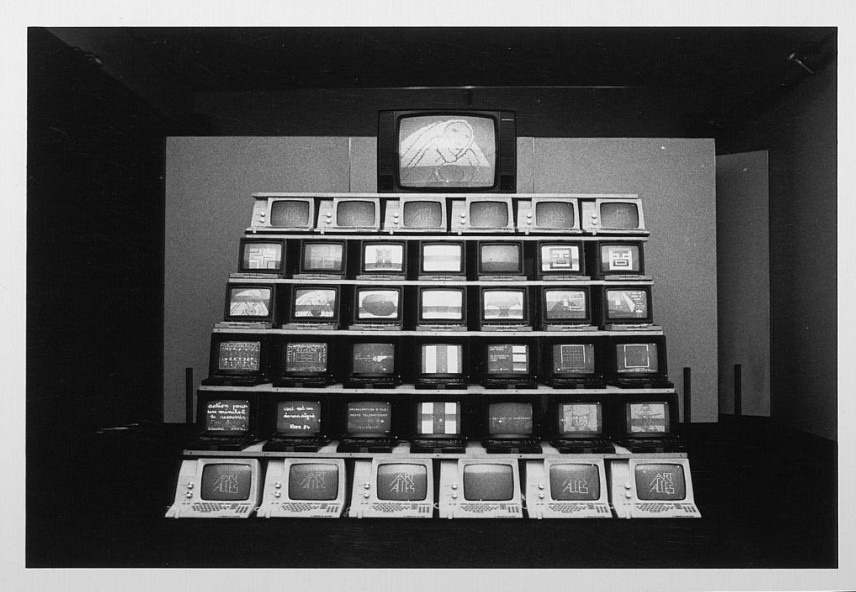

**Figure 10.** Installation of *Art Accès* at the Locarno International Video Festival in 1985. Collection Frédéric Develay © Kandinsky Library, MNAM/CCI, Centre Pompidou.

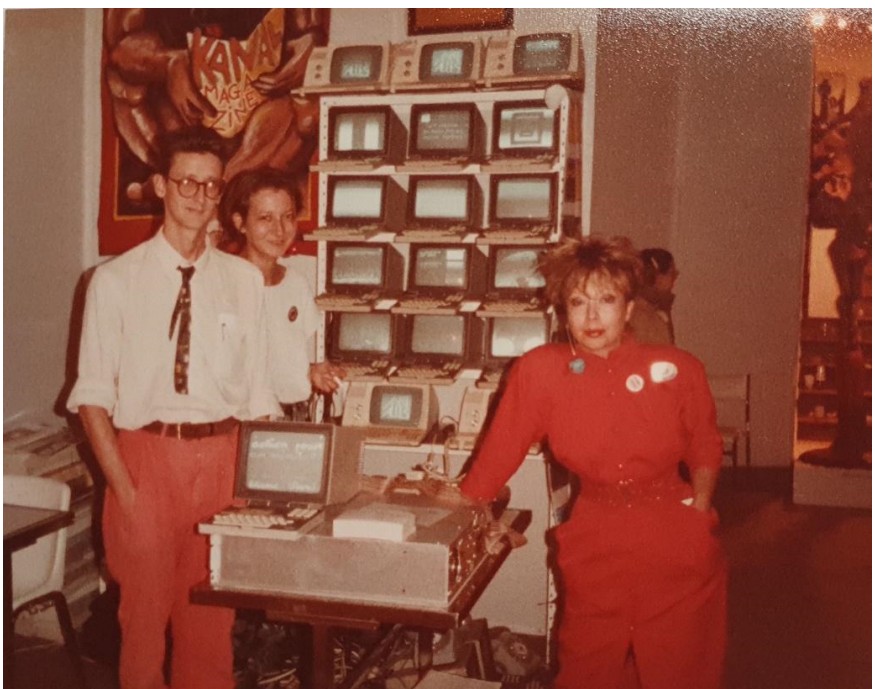

**Figure 11.** Installation of *Art Accès* at the 1985 FIAC with Frédéric Develay and ORLAN. Collection Frédéric Develay © Kandinsky Library, MNAM/CCI, Centre Pompidou.

### 6. The Limits of the Experiment

If the magazine disappeared after three years, it was largely because it failed to create the enthusiasm its creators had hoped for.[18] Apart from the art world, it attracted only a small audience (See (Anis and Lebrave 1987, p. 71)). Minitel has therefore not been the great tool for cultural democratisation that was imagined. "Obviously, *ART ACCES Revue* also had the same hope, carried on the Internet later on and with video at the time, of a democratisation of art, by by-passing the cultural institutions and intermediaries in order to reach the audience directly at home—and based on a new economic model that was considered sustainable" (Bureaud 2016, p. 144). Annick Bureaud notes that it was a failure in all three cases. The lack of interactivity in the projects proposed by the artists and the limited use of messaging, which would have allowed for a real dialogue between the artists and the users of the service, were among the criticisms levelled at this telematic project (Anis and Lebrave 1987, pp. 71–72). Despite its name as a magazine ("revue" in French), *Art Accès* was closer to an online gallery or a creative platform than to an art magazine that would function by issue. From this unique pre-Internet artistic experience, we can nevertheless remember the exceptional number of works created especially for the Minitel by the most important artists of the time in their respective fields (including Roy Ascott, Daniel Buren, John Cage, John Giorno, Jenny Holzer, Lea Lublin, Vera Molnar, and many others), each of whom experimented and played with the videotex. In total, more than 1500 screen pages were created within the framework of *Art Accès* (Develay and ORLAN 1987, p. 57) of which unfortunately few traces remain today. In-depth research is still needed to reconstruct the history of this telematic adventure.

**Funding:** This research received the "Support for research in art theory and criticism" from the Centre national des arts plastiques (National Centre for Visual Arts, France).

**Conflicts of Interest:** The author declares no conflict of interest.

### Notes

[1]   The word "telematic" refers in French only to the Minitel, but it has been used more widely in English to describe early network art, which precisely couples telecommunication and informatics. See (Bureaud 2016, p. 139).

[2]   When the network was still functioning, it was not possible to save data either, since the Minitel was only an access and consultation terminal without any storage capacity, unlike a computer terminal.

[3]   The archive collection, which gives only a very incomplete account of the magazine's content, includes photographs, slides, photocopies of thermal prints, correspondence between the artists and the magazine's creators, preparatory drawings, texts and paper documentation.

[4]   The impetus for the development of the Minitel was the report that Simon Nora and Alain Minc submitted to the President of the French Republic in 1978. See (Nora and Minc 1978).

[5]   Cost was an important but not the only barrier. In the United Kingdom and Germany, the number of services available was very limited, which did not encourage private individuals to equip themselves with a terminal, while in the United States, no major national equipment project was organised. See (Schafer and Thierry 2012, pp. 37–43).

[6]   The Minitel 1 Colour is offered for rental and maintenance from May 1985 for 200 francs per month, unlike the black and white terminal which is distributed free of charge to every household. See (Rizzo-Vignaud n.d.).

[7]   ORLAN, *Étude documentaire: Le Drapé-Le Baroque, Sein Unique, Monstration Phallique*, 1983, black and white photograph, 100 × 100 cm.

[8]   See the note by François Morellet, Collection Frédéric Develay, Kandinsky Library, MNAM/CCI, Centre Pompidou.

[9]   François Morellet, *16 carrés*, 1953, oil on wood, 80 × 80 cm, Mönchengladbach, Städtisches Museum Abteiberg.

[10]  See the work of François Morellet, *Reflets dans l'eau déformés par le spectateur*, 1964, wood, plywood, white neon tubes, metal tray, water, manual mechanical system, Ivry-sur-Seine, MAC VAL.

[11]  François Morellet, *Trame 5° placée horizontalement*, 1976, acrylic on canvas, 200 × 200 cm, Berlin, Staatliche Museen Preussischer Kulturbesitz, Nationalgalerie.

[12]  François Morellet, *Géométree n° 92*, 1985, wood and acrylic, 200 × 200 cm, Paris, collection SJB.

[13]  See (Develay and ORLAN 1987, p. 57). The magazine was distributed at the same time by the MIRABEL server of the City of Metz which hosted it. During the years of existence of *Art Accès*, the magazine was hosted on different servers, including that of the City of Metz and those of the French newspaper *Libération* and of the news magazine *Le Nouvel Observateur*.

14    The magazine was presented in the section called "Mémoires artificielles" ["Artificial Memories"] along with two other projects: the videodisc *Iconographie* and the telematic project *Écran du livre*. See the page of "Mémoires artificielles", in (Chaput and Lyotard 1985).

15    The sixth edition of the Videoart festival took place from 3 to 7 August 1985.

16    It should be noted that the French Télétel network was not accessible from Switzerland. This type of installation therefore made it possible to present the magazine during the festival. Thanks to a "Christel" memory box, it was also possible to scroll through some of the magazine's creations on a Minitel terminal present in the room. See https://videoartfestival.ch/darchive/vaf-1985-presse-vie-festival-video-masi/, accessed 20 May 2022.

17    The FIAC was held from 5 to 13 October 1985.

18    The reasons for stopping *Art Accès* are multiple. The economic model of the magazine was not sustainable in the long term. *Art Accès* was under the status of a non-profit association, so the founders did not make any money from it, while the hosting and the technical production of the magazine required money. For the companies hosting the magazine, the project was not interesting enough because it did not bring them enough connections. Email from Frédéric Develay to the author, 22 October 2022.

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
