# Peer review of "The French Telematic Magazine Art Accès (1984–1987)"

_arts, 1984_

Round 1

Reviewer 1 Report

This article proposes an initial study of the magazine Art Accès, produced for the medium of the Minitel between 1984 and 1987. The text gives us an encompassing context of the publication and analyses some of the works for which some documentation still exists.

The initial summary and contextualisation are very complete. The one issue I find there is that from the descriptions that are given of the Minitel, it doesn’t become quite clear whether this system consisted only of videotext or if the protocols supported different formats. Maybe some more detail in the technical description of the system would help, especially related to technical terminology that may be obsolete or less common nowadays, such as e.g. “alpha mosaic mode”, “frames”, or “masking or dynamic formatting”. Actually, it would have been very interesting to have access to the four pages of instructions that were given to participants and that are mentioned in line 103.

The description of the visual works is quite interesting, connecting several of the works to which some documentation exists and, while not managing to develop a very broad critical analysis or to focus on the varieties of aesthetic experience or of creative approaches in much detail, it is nevertheless an interesting and informative read.

The section on literature is quite the opposite, extremely paired down, briefly mentioning hypertext and experimental literacy forms but not really developing this.

As for music, the section is extremely brief too, and points to the development of experimental systems of notation, depicting one, but unfortunately not expanding this much (though John Cage is mentioned in Figure 2, his work is not further described)

The discussion about the presence of works in exhibitions is quite interesting. 

The conclusion is very frank regarding the ultimate failure of the project and points to some of the reasons behind it without however discussing it in depth (not mentioning aspects related to the business model of the magazine, its funding or economics, etc.).

This is a very compelling read and a good contribution to a study that can and should be further developed. Let’s hope this work can be done by the authors or by other researchers. As the authors say on line 340, Art Accès had contributions by some of “the most important artists of the time in their respective fields” and it would be wonderful to be able to recover and study them (and maybe even preserve the works somehow).

Minor notes and suggestions for edits:

  • Line 27: “It was seen by many as the forerunner of the Internet as we know it today.” — I think it would be clearer to phrase this as “one of the forerunners”. Also, it would be helpful to include at least one reference here.
  • Line 34: Mintel > Minitel
  • Line 61: “This initiative sets France apart from other countries with a comparable service where users paid for the rental of the terminal and where use was largely reserved for professionals.” — Some references would be helpful here.
  •  

Author Response

Thank you very much for your review. It was very hepful for me to have such precise and constructive feedback. I have taken your comments into account and tried to clarify things where I can, but I am at the beginning of my research and there are still many unanswered questions.

About the system of the Minitel : It was only a videotex system like I wrote in the first paragraph of my article. As you suggested, I have added the four pages of instructions to the artists in my article.

For the literary section, I have very little information. So I couldn't go any further. This is an area to be explored in the future. For the music section, I don't have much information either but I added an example to complete this part.

I have added a note on the reasons for stopping the magazine : "The reasons for stopping Art Accès are multiple. The economic model of the magazine was not sustainable in the long term. Art Accès was under the status of a non-profit association, so the founders did not make any money from it, while the hosting and the technical production of the magazine required money. For the companies hosting the magazine, the project was not interesting enough because it did not bring them enough connections."

Thank you again for your remarks. This is really encouraging for my further research on this subject.
I am sending you my revised article.

All my best,

MV

Reviewer 2 Report

Terrific analysis and research of a fascinating topic that most readers will have previously known very little about 

Author Response

Thank you very much for your review. It is very encouraging to read such a positive review when there is still a lot of research to be done on this subject.

I am sending you the revised version of my article.

All my best,

MV
